# Aortic Agatston score correlates with the progression of acute type A aortic dissection

**Yasushi Tashima**[1,2]*, **Shinichi Iwakoshi**[3], **Takeshi Inoue**[3], **Noriyuki Nakamura**[1,2], **Taichi Sano**[1,2], **Naoyuki Kimura**[2], **Takashi Inoue**[4], **Koichi Adachi**[1,2], **Atsushi Yamaguchi**[2]

**1** Department of Cardiovascular Surgery, Yokosuka General Hospital Uwamachi, Yokosuka, Kanagawa, Japan, **2** Department of Cardiovascular Surgery, Saitama Medical Center, Jichi Medical University, Saitama, Japan, **3** Department of Radiology, Nara Medical University, Nara, Japan, **4** Institute for Clinical and Translational Science, Nara Medical University, Kashihara, Japan

* tyasushi42@gmail.com

**Data Availability Statement:** Data cannot be shared publicly because of patients privacy. Data are available from the Yokosuka General Hospital Uwamachi ethics committee (contact via TEL: +8146-823-2630, FAX:+8146-827-1305) for

## Abstract

Aortic calcification in the tunica media is correlated with aortic stiffness, elastin degradation, and wall shear stress. The study aim was to determine if aortic calcifications influence disease progression in patients with acute type A aortic dissection (ATAAD). We retrospectively reviewed a total of 103 consecutive patients who had undergone surgery for ATAAD at our institution between January 2009 and December 2019. Of these, 85 patients who had preoperatively undergone plain computed tomography angiography (CTA) for evaluation of their aortic calcification were included. Moreover, we assessed the progression of aortic dissection after surgery via postoperative CTA. Using a classification and regression tree to identify aortic Agatston score thresholds predictive of disease progression, the patients were classified into high-score (Agatston score $\geq$ 3344; n = 36) and low-score (<3344; n = 49) groups. Correlations between aortic Agatston scores and CTA variables were assessed. Higher aortic Agatston scores were significantly correlated with the smaller distal extent of aortic dissection ($p$ < 0.001), larger true lumen areas of the ascending ($p$ = 0.009) and descending aorta ($p$ = 0.002), and smaller false lumen areas of the descending aorta ($p$ = 0.028). Patients in the high-score group were more likely to have DeBakey type II dissection ($p$ = 0.001) and false lumen thrombosis ($p$ = 0.027) than those in the low-score group, thereby confirming the correlations. Aortic dissection in the high-score group was significantly less distally extended ($p$ < 0.001). A higher aortic Agatston score correlates with the larger true lumen area of the ascending and descending aorta and the less distal progression of aortic dissection in patients with ATAAD. Interestingly, the findings before and after surgery were consistent. Hence, aortic Agatston scores are associated with aortic dissection progression and may help predict postoperative residual dissected aorta remodeling.

## Introduction

Aging, dyslipidemia, tobacco use, inflammatory disease, chronic kidney disease, and diabetes mellitus are considered to be factors that predispose individuals to aortic calcification, which is

researchers who meet the criteria for access to confidential data.

**Funding:** The author(s) received no specific funding for this work.

**Competing interests:** The authors have declared that no competing interests exist.

increasingly recognized as a strong predictor of cardiovascular events and all-cause mortality [1–3]. Interestingly, aortic calcification reportedly can be detected in the tunica media of the human aorta before being observed in neo-intima plaques [4, 5]. Moreover, pathological research has indicated that the average amount of calcification at all ages is higher in the tunica media than in the intima [6, 7].

ATAAD is a life-threatening cardiovascular event that requires immediate surgical repair [8]. Aortic dissection results from the separation of the aortic wall layers, and a tear in the intimal layer allows blood to enter the tunica media, which causes progressive dissection [9]. Aortic calcifications extending from the aortic intima to the media may prevent progression of aortic dissection by restricting the separation [10]. Although several studies have investigated the association between aortic calcifications and abdominal aortic aneurysms (AAAs) [11], few have examined the association of aortic calcifications with ATAAD [12–14]. Aortic calcifications have been shown to increase the peak wall shear stress and decrease the biomechanical stability of AAAs [15]. Furthermore, aortic stiffening, increased pulse pressure, reduced coronary blood flow, and left ventricular hypertrophy have been found to be strongly associated with aortic calcifications in the intima and media [16–18]. It is plausible that aortic calcifications could alter the biomechanical properties of the aorta in patients with acute type A aortic dissection (ATAAD), thereby influencing aortic dissection progression.

The severity of coronary artery calcification is often represented by an Agatston score, which is an independent risk marker for cardiac events, cardiac mortality, and all-cause mortality [19, 20]. Even though aortic Agatston scores have been used in a large number of studies for assessment of aortic calcification severity [21–24], no study to date has focused on aortic Agatston scores in patients with ATAAD. Hence, we conducted this study to explore the association of aortic Agatston scores with disease progression in patients with ATAAD.

## Materials and methods

### Patients

After obtaining approval and a waiver of informed consent from our Institutional Review Board, we retrospectively reviewed a total of 103 consecutive patients who had undergone emergency surgery for ATAAD at Yokosuka General Hospital Uwamachi between 2009 and 2019. Eighteen patients without preoperative aortic calcification measurements by plain computed tomography angiography (CTA) were excluded. The remaining 85 patients were included in the study. We used a classification and regression tree (CART), a machine-learning algorithm for clinical decision-making that can be used to determine the breakpoint and identify aortic Agatston score thresholds predictive of disease progression (distal extent of aortic dissection) in patients with ATAAD (S1 Fig). CART is a nonparametric decision tree learning technique that produces either classification or regression trees based on whether the dependent variable is categorical or numeric, respectively [25, 26]. The patients were divided into two groups according to the Agatston score cutoff value identified by CART (cutoff value = 3344): the high-score group had 36 patients with Agatston score ≥ 3344, whereas the low-score group had 49 patients with Agatston score < 3344. Then, we examined the association between aortic calcifications and the extent of aortic dissection in these patient groups. Furthermore, to investigate the early progression of aortic dissection after surgery, we also examined the association between aortic calcifications and postoperative CTA variable of the descending aorta in the patients, excluding those who had DeBakey type II dissection (n = 20) and those who died before performing the postoperative CTA (n = 4). The patient selection flowchart is illustrated in Fig 1.

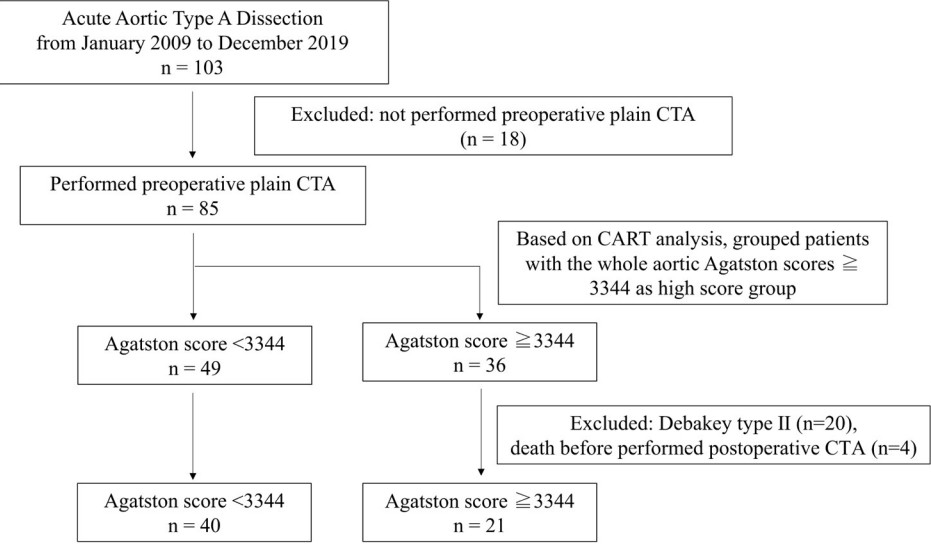

CTA = computed tomography angiography, CART = classification and regression tree

**Fig 1. Patient selection flowchart.**

## Data collection

CTA and echocardiography were performed to establish a definitive diagnosis. Upon confirmation of the ATAAD diagnosis, the patients were transferred to the operating room as soon as possible. Intraoperative findings confirmed ATAAD as well. Postoperative CTA was performed within 7 days after surgery to assess early aortic dissection progression after the initial surgery and changes in CTA variables from pre- to post-surgery. Furthermore, postoperative plain computed tomography (CT) scan was conducted 6 months after surgery to assess midterm changes in descending aortic dimension and diameter after surgery between the two groups. Data on the following variables were collected from the patients' medical records and compared between the two groups: preoperative CTA variables, including aortic diameters, area of the ascending aorta at the level of the right pulmonary artery, area of the descending aorta at the level of the aortic valve, true and false lumen areas, the location of major entry, distal extent of aortic dissection, volume and surface area of total aortic calcifications, and aortic Agatston score; postoperative CTA variables, including the totally thrombosed false lumen of both the descending and abdominal aortas, aortic diameters, total aortic area, and true and false lumen areas of the descending aorta at the level of the aortic valve. The location of the major entry was identified on the preoperative CTA, and it was confirmed during surgery in case the entry site was located in the proximal aorta (aortic root, ascending aorta) or the aortic arch.

## Measurement of aortic calcifications and areas on CTA

Aortic calcifications were measured by using Synapse Vincent software (version 5.3; Fujifilm, Tokyo, Japan). Calcified lesions located from the sinuses of Valsalva to the aortic bifurcation were identified with a density of $> 130$ Hounsfeld units (HU) on preoperative plain CT. Subsequently, a calcium score was calculated for each region by multiplying the area by a cofactor (i.e., cofactor 1, 130–199 HU; cofactor 2, 200–299 HU; cofactor 3, 300–399 HU; and cofactor 4, $> 400$ HU). Finally, a total aortic Agatston score was calculated by adding the scores for all

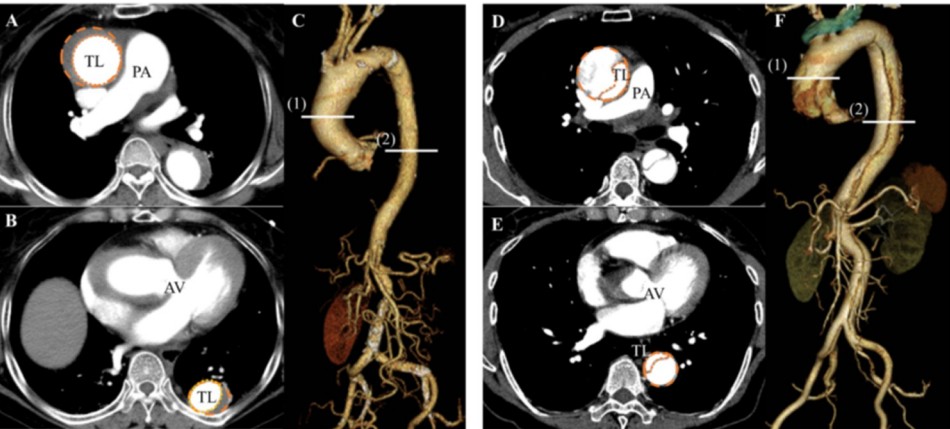

**Fig 2. Representative images of aortic area measurements.** (A), (B), and (C) depict representative computed tomography images from patients in the high-score group, whereas (D), (E), and (F) are those from patients in the control group. (C), (F) Total and true aortic lumen areas were evaluated at the (1) ascending and (2) descending aorta. (A), (D) The delineated area on the ascending aorta was measured at the level of the right pulmonary artery. (B), (E) The delineated area on the descending aorta was assessed at the level of the aortic valve. Total and true aortic lumen areas are marked by broken-line and dotted-line circles, respectively. AV = aortic valve; Asc = ascending aorta; Des = descending aorta; PA = pulmonary artery; TL = true lumen.

individual lesions. CT images were reconstructed at a 5-mm slice thickness on a Phillips Brilliance CT64 (Philips, Amsterdam, Netherlands) or TSX301B-1A (Canon, Tokyo, Japan).

As false and true lumen areas, the ratio of false lumen area to true lumen area, and false lumen thrombosis are associated with disease prognosis, we also measured these variables to assess their association with aortic calcification. Total aortic area and diameters as well as true and false lumen areas were measured for the ascending aorta at the level of the right pulmonary artery and for the descending aorta at the aortic valve level. These parameters, however, were not evaluated for the descending aorta in patients with DeBakey type II dissection. Representative images for both groups are shown in Fig 2. The area ratio of the true lumen to total lumen was calculated as follows:

$$True\ lumen/total\ lumen\ area\ ratio = \frac{True\ lumen\ area}{Total\ aortic\ lumen\ area}$$

The false lumen area was calculated as the true lumen area subtracted from the total aortic lumen area.

## Assessment of disease progression in aortic dissection on CTA

The Society for Vascular Surgery/Society of Thoracic Surgeons (SVS/STS) Aortic Dissection Classification System was used to assess the progression of aortic dissection [27]. According to the SVS/STS system, the distal extent score defined the zone to where the aortic dissection was distally extended to as shown in Fig 3. The distal extent score ranged from 0 (aortic dissection within ascending aorta) to 12 (aortic dissection extended to femoral artery), wherein lower scores reflected less progression of the dissection.

## Surgical procedure

Our surgical procedure consisted of a median sternotomy with a standard cardiopulmonary bypass. The subclavian artery, left ventricular apex, or femoral artery was used for arterial cannulation. An antegrade or retrograde infusion of cold blood cardioplegic solution was

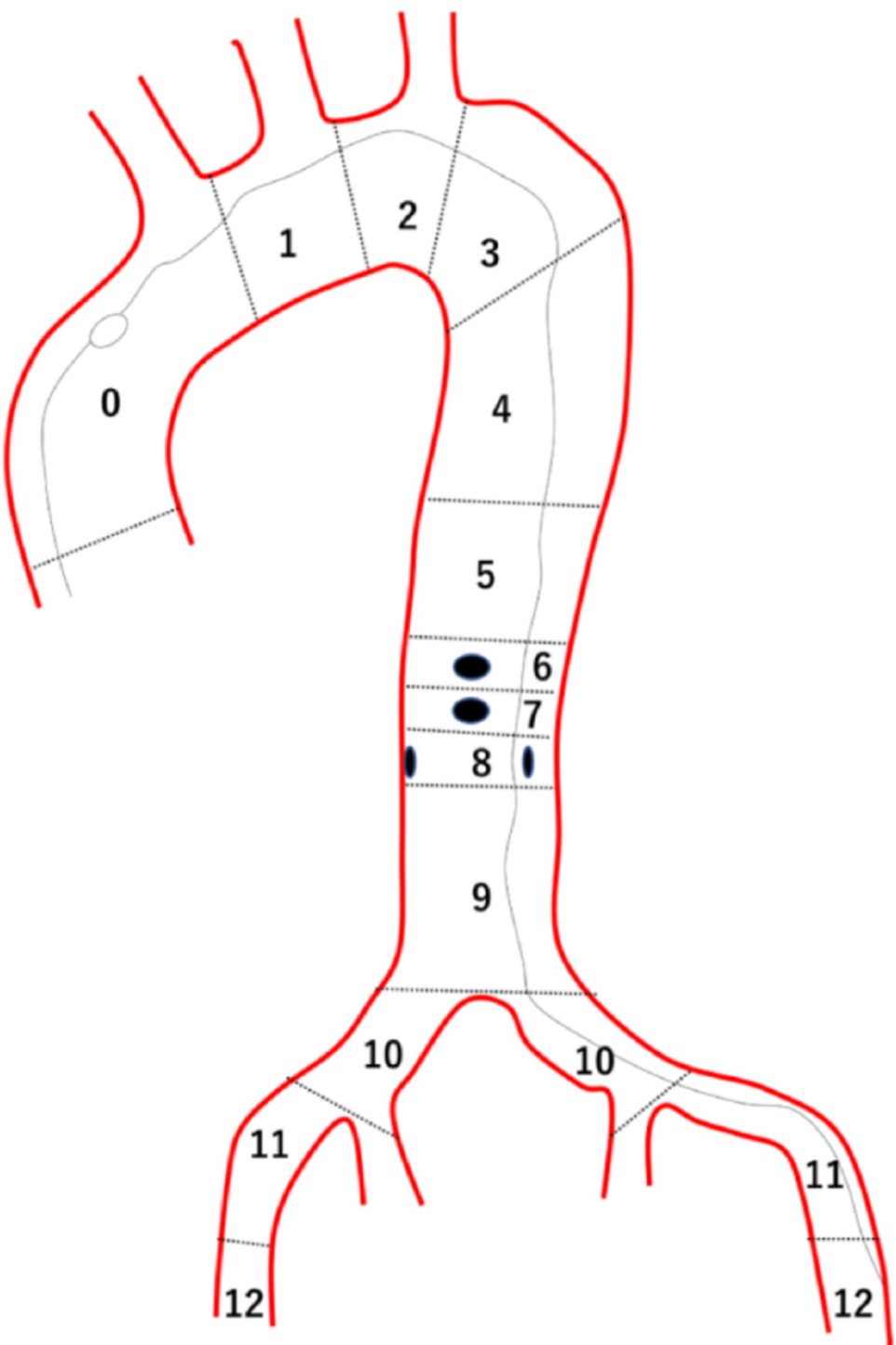

**Fig 3. Scheme of distal extent score according to the Society for Vascular Surgery/Society of Thoracic Surgeons Aortic Dissection Classification System. A.** In the example illustrated, the dissection process extends distally to zone 12, which indicates the distal extent score of "12".

administered for myocardial protection. Surgery was performed under hypothermic circulatory arrest (bladder temperature, 20˚C–26˚C), and open distal anastomosis was performed under circulatory arrest with or without antegrade selective cerebral perfusion. Basically, a

tear-oriented surgical strategy was adopted [28]. An ascending aortic replacement was performed when the entry site was located in the ascending aorta or when an entry tear could not be identified in the ascending aorta or aortic arch (DeBakey IIIb retrograde dissection). On the other hand, total or partial arch replacement was performed for patients with an entry site in the aortic arch. Although aortic valves were preserved whenever possible, we performed aortic root replacement for cases with an intimal tear extending to the sinuses of Valsalva or those with aortic root dilation associated with annuloaortic ectasia.

## Statistical analysis

Continuous data were expressed as the median (interquartile range) and compared between the two groups by performing the Mann–Whitney U test. Categorical data were expressed as frequencies (%) and analyzed by performing the chi-square test or Fisher's exact test. To examine the relationship between aortic Agatston scores and CTA variables, correlation coefficients were calculated by using the nonparametric Spearman correlation analysis. All statistical analyses were performed by using EZR software (Saitama Medical Center, Jichi Medical University, Saitama, Japan). Values of $p < 0.05$ were considered to be indicative of statistical significance.

## Results

### Patients' characteristics

The preoperative clinical characteristics of the patients are presented in Table 1. The patients in the high-score group were significantly older than those in the low-score group (77.5 vs. 63

**Table 1. Preoperative characteristics.**

| Characteristics | Total (n = 85) | Low-score group (n = 49) | High-score group (n = 36) | p-value |
|---|---|---|---|---|
| Age | 68 (60–77) | 63 (53–69) | 77.5 (71–83) | <0.001 |
| Female | 44 (51.8%) | 20 (40.8%) | 24 (66.7%) | 0.028 |
| BMI | 23.8 (21–27.6) | 25.3 (21.3–29.8) | 22.8 (19.5–24.5) | 0.002 |
| Hypertension | 73 (85.9%) | 40 (81.6%) | 33 (91.7%) | 0.224 |
| Diabetes mellitus | 6 (7.1%) | 5 (10.2%) | 1 (2.8%) | 0.236 |
| Chronic kidney disease | 46 (54.1%) | 23 (46.9%) | 23 (63.9%) | 0.131 |
| Hyperlipidemia | 20 (23.5%) | 11 (22.4%) | 9 (25%) | 0.801 |
| Ischemic heart disease | 3 (3.5%) | 1 (2%) | 2 (5.6%) | 0.571 |
| Peripheral artery disease | 0 (0%) | 0 (0%) | (0%) | >0.99 |
| Smoking history | 20 (23.5%) | 15 (30.6%) | 5 (13.9%) | 0.119 |
| Preoperative shock | 12 (14.1%) | 4 (8.2%) | 8 (22.2%) | 0.112 |
| Cardiac tamponade | 46 (54.1%) | 20 (40.8%) | 26 (72.2%) | 0.005 |
| Cardiopulmonary resuscitation | 2 (2.4%) | 2 (4.1%) | 0 (0%) | 0.506 |
| Neurologic deficit | 5 (5.9%) | 3 (6.1%) | 2 (5.6%) | >0.99 |
| Malperfusion | | | | |
| Paraplegia | 2 (2.4%) | 2 (4.1%) | 0 (0%) | 0.506 |
| Limb | 11 (12.9%) | 7 (14.3%) | 4 (11.1%) | 0.753 |
| Renal | 9 (10.6%) | 6 (12.2%) | 3 (8.3%) | 0.727 |
| Brain | 11 (12.9%) | 8 (16.3%) | 3 (8.3%) | 0.342 |
| Coronary | 3 (3.5%) | 1 (2%) | 2 (5.6) | 0.571 |
| Mesenteric | 11 (12.9%) | 9 (18.4%) | 2 (5.6%) | 0.108 |

BMI, body mass index.

years, respectively; $p < 0.001$). Furthermore, the body mass index was significantly lower in the high-score group than in the low-score group ($p = 0.002$). There were no significant differences in sex, hypertension, diabetes mellitus, chronic kidney disease, hyperlipidemia, ischemic heart disease, peripheral artery disease, smoking history, preoperative shock status (systolic blood pressure < 80 mmHg), and neurological deficit between the two groups. Interestingly, cardiac tamponade was more frequently observed in the high-score group than in the low-score group ($p = 0.005$). No significant difference was observed in each malperfusion between the studied groups.

## Correlations between Agatston scores and preoperative CTA variables

Spearman correlation coefficients were used to evaluate the relationships between Agatston scores and preoperative CTA variables (Table 2). The Agatston scores were highly correlated with the average CT value ($p < 0.001$), maximum CT value ($p < 0.001$), aortic calcification volume ($p < 0.001$), and aortic calcification surface area ($p < 0.001$). Despite having no correlations with the diameters and total areas of the ascending and descending aortas, the Agatston scores were significantly correlated with the true lumen areas of the ascending and descending aorta ($p = 0.009$ and $p = 0.002$, respectively) and with the ratios of the true lumen area to total lumen area for the ascending and descending aortas ($p = 0.009$ and $p < 0.001$, respectively). Although the Agatston scores did not correlate with the false lumen area of the ascending aorta, they displayed a significant correlation with the false lumen area of the descending aorta ($p = 0.028$). The results suggested that higher Agatston scores were significantly correlated with larger true lumen areas of the ascending and descending aortas and with smaller false lumen areas of the descending aorta. Interestingly, the correlation of Agatston scores with the true and false lumen areas of the descending aorta appeared to be stronger than the correlation with the true and false lumen areas of the ascending aorta.

As shown in Table 3, DeBakey type II dissection and false lumen thrombosis were more commonly observed in the high-score group than in the low-score group ($p = 0.036$, $p = 0.002$, respectively). Further, the distal extent score was significantly lower in the high-score group

**Table 2. Correlation between Agatston score and CTA variables.**

| Correlations | rho | *p*-value |
|---|---|---|
| Distal extent score | −0.494 | <0.001 |
| Asc diameter (mm) | 0.184 | 0.094 |
| Asc area (mm$^2$) | 0.203 | 0.062 |
| True lumen area of Asc (mm$^2$) | 0.28 | 0.009 |
| False lumen area of Asc (mm$^2$) | −0.103 | 0.357 |
| True lumen/total lumen area ratio of Asc | 0.28 | 0.009 |
| Des diameter (mm) | 0.057 | 0.647 |
| Des area (mm$^2$) | 0.081 | 0.522 |
| True lumen area of Des (mm$^2$) | 0.456 | 0.002 |
| False lumen area of Des (mm$^2$) | −0.273 | 0.028 |
| True lumen/total lumen area ratio of Des | 0.487 | <0.001 |
| Average of CT value | 0.795 | <0.001 |
| Max of CT value | 0.879 | <0.001 |
| Calcification volume (mm$^3$) | 0.996 | <0.001 |
| Calcification surface area (mm$^2$) | 0.998 | <0.001 |

CTA, computed tomography angiography; Asc, ascending aorta; Des, descending aorta.

**Table 3. Preoperative CTA variables.**

| Preoperative CTA variables | Total (n = 85) | Low-score group (n = 49) | High-score group (n = 36) | *p*-value |
|---|---|---|---|---|
| DeBakey I or IIIb retrograde | 65 (76.5%) | 44 (89.8%) | 21 (58.3%) | 0.001 |
| DeBakey II | 20 (23.5%) | 5 (10.2%) | 15 (41.7%) | 0.001 |
| Distal extent score | 8 (4–10) | 10 (7–11) | 5 (0–8) | <0.001 |
| Major entry location | | | | |
| Aortic root | 5 (5.9%) | 2 (4.1%) | 3 (8.3%) | 0.646 |
| Ascending aorta | 43 (50.6%) | 25 (51%) | 18 (50%) | >0.99 |
| Aortic arch | 24 (28.2%) | 16 (32.7%) | 8 (22.2%) | 0.337 |
| Descending aorta | 2 (2.4%) | 1 (2%) | 1 (2.8%) | >0.99 |
| Unidentified | 11 (12.9%) | 5 (10.2%) | 6 (16.7%) | 0.516 |
| False lumen thrombosis | 39 (45.9%) | 17 (34.7%) | 22 (61.1%) | 0.027 |
| Asc diameter (mm) | 46.5 (43.6–50.1) | 46.5 (43.3–48.8) | 47.2 (44.2–51.3) | 0.268 |
| Asc area (mm$^2$) | 1752 (1554–2026) | 1745 (1544–1996) | 1794 (1609–2160) | 0.185 |
| True lumen area of Asc (mm$^2$) | 581 (333–993) | 497 (320–782) | 743 (376–1150) | 0.033 |
| False lumen area of Asc (mm$^2$) | 1110 (795–1404) | 1119 (833–1442) | 1083 (761–1295) | 0.388 |
| True lumen/total lumen area ratio of Asc | 0.34 (0.21–0.52) | 0.28 (0.2–0.47) | 0.38 (0.23–0.6) | 0.058 |
| Des diameter (mm) | 30.9 (29.7–34.1) | 31 (29.5–33.5) | 30.7 (29.9–34.3) | 0.874 |
| Des area (mm$^2$) | 800 (688–906) | 805 (681–890) | 763 (698–919) | 0.833 |
| True lumen area of Des (mm$^2$) | 295 (229–393) | 256 (207–351) | 385 (335–430) | 0.002 |
| False lumen area of Des (mm$^2$) | 483 (406–593) | 518 (441–612) | 413 (337–533) | 0.017 |
| True lumen/total lumen area ratio of Des | 0.36 (0.29–0.48) | 0.32 (0.28–0.39) | 0.47 (0.38–0.58) | 0.001 |
| Average of CT value | 273 (238–314) | 245 (212–269) | 314 (298–365) | <0.001 |
| Max of CT value | 1026 (678–1507) | 730 (439–986) | 1600 (1264–1746) | <0.001 |
| Calcification volume (mm$^3$) | 3627 (519–9892) | 629 (328–2722) | 12080 (8034–20373) | <0.001 |
| Calcification surface area (mm$^2$) | 629 (104–1978) | 122 (57–475) | 2416 (1607–4075) | <0.001 |
| Agatston score | 2171 (352–7519) | 381 (162–1574) | 8718 (5374–14939) | <0.001 |

CTA, computed tomography angiography; Asc, ascending aorta; Des, descending aorta.

than in the low-score group (5 vs. 10, *p* < 0.001). No significant difference was found in the major entry location. The true lumen areas of the ascending and descending aortas were significantly larger in the high-score group than in the low-score group (731.9 vs. 480.4 mm$^2$, *p* = 0.025; 376.2 vs. 250.9 mm$^2$, *p* < 0.001), whereas the diameters and total areas of the ascending and descending aortas were not significantly different between the groups. We also found that the true lumen area/total lumen area ratios of the ascending and descending aortas were significantly higher in the high-score group than in the low-score group (0.369 vs. 0.293, *p* = 0.039; 0.469 vs. 0.323, *p* < 0.001). Furthermore, the false lumen area of the descending aorta was significantly smaller in the high-score group than in the low-score group (421.9 vs. 519.5 mm$^2$, *p* = 0.02). Additionally, we noticed that the average CT value, maximum CT value, aortic calcification volume, and aortic calcification surface area were significantly greater in the high-score group than in the low-score group (*p* < 0.001 for all variables).

## Associations between Agatston scores and postoperative CTA variables in patients with DeBakey I or IIIb retrograde

To investigate the association between aortic calcification and early remodeling of aortic dissection after surgery among the 61 patients with DeBakey I or IIIb retrograde, postoperative CTA variables were compared between 21 patients in the high-score group and 40 patients in

**Table 4. Surgical procedures and postoperative CTA variables in the patients with DeBakey I or IIIb retrograde.**

| Postoperative CTA variables | Total (n = 61) | Low-score group (n = 40) | High-score group (n = 21) | *p*-value |
|---|---|---|---|---|
| Ascending aorta replacement | 49 (80.3%) | 31 (77.5%) | 18 (85.7%) | 0.518 |
| Aortic arch replacement | 10 (16.4%) | 8 (20%) | 2 (9.5%) | 0.47 |
| Aortic root replacement | 2 (3.3%) | 1 (2.5%) | 1 (4.8%) | >0.99 |
| Concomitant AVR | 5 (8.2%) | 3 (7.5%) | (9.5%) | >0.99 |
| Concomitant CABG | 1 (1.6%) | 0 (0%) | 1 (4.8%) | 0.344 |
| False lumen thrombosis | 38 (62.3%) | 19 (47.5%) | 17 (81%) | 0.015 |
| Distal extent score | 9 (8–11) | 10 (9–11) | 8 (6–10) | 0.025 |
| Des diameter (mm) | 32.3 (30.8–35.3) | 32.3 (31.1–34.4) | 33.2 (30.6–36.3) | 0.606 |
| Des area (mm$^2$) | 841 (748–960) | 819 (760–895) | 859 (715–1077) | 0.495 |
| True lumen area of Des (mm$^2$) | 379 (261–515) | 315 (240–482) | 486 (377–663) | 0.002 |
| False lumen area of Des (mm$^2$) | 451 (327–572) | 496 (394–582) | 327 (276–551) | 0.024 |
| True lumen/total lumen area ratio of Des | 0.47 (0.31–0.6) | 0.4 (0.29–0.53) | 0.6 (0.47–0.67) | 0.003 |

CTA, computed tomography angiography; AVR, aortic valve replacement; CABG, coronary artery bypass grafting; Des, descending aorta.

the low-score group (Table 4). No significant difference was found in the surgical procedures between the two groups. The distal extent score after surgery was significantly lower in the high-score group than in the low-score group ($p < 0.001$). The false lumens of both the descending and abdominal aortas were more frequently totally thrombosed in the high-score group after surgery than in the low-score group (75% [21 of 28] vs. 32.4% [12 of 37], respectively; $p = 0.001$). Although the aortic diameter and total aortic area of the descending aorta were not significantly different between the groups, the true lumen area and true lumen/total lumen area ratio of the descending aorta were significantly larger in the high-score group than in the low-score group (486 vs. 301 mm$^2$, $p = 0.001$; 0.6 vs. 0.41, $p = 0.003$, respectively). Further, the false lumen area of the descending aorta was significantly smaller in the high-score group than in the low-score group ($p = 0.042$). Compared with preoperative CTA variables, the postoperative true lumen dimension of the descending aorta was small in the high- and low-score groups (p = 0.038, 0.058, respectively). However, the distal extent score after surgery did not significantly change (S1 Table). Moreover, there was no remarkable difference in the changes in the distal extent score and true and false lumen area of the descending aorta from pre- to post-surgery between the two groups (S2 Table). Hence, high aortic Agatston scores could be correlated with a slower progression of residual dissected descending aorta before and after surgery.

Moreover, we assessed and compared the midterm changes in descending aortic diameter and area via plain CT scan 6 months after surgery and CT variables within 7 days after surgery. Interestingly, dilatation of the descending aortic diameter and area 6 months after surgery were smaller in the high-score group than in the low-score group (Des diameter change: 0.98 [0.87–1.04] vs. 1.01 [0.94–1.1], p = 0.058; Des area change: 1.03 [0.88–1.13] vs. 1.11 [0.97–1.29], p = 0.031, in fold change/CT variables within 7days after surgery) (S3 Table).

## Discussion

This study demonstrated the following: 1) aortic Agatston scores significantly correlated with the progression of aortic dissection and the true lumen areas and true lumen area/total lumen area ratios of the ascending and descending aortas; 2) DeBakey type II dissection and false lumen thrombosis were significantly more likely and aortic dissection was less distally extended in the high-score group than in the low-score group; and 3) consistent with the

preoperative findings, in the postoperative CTA variables among the patients with DeBakey I or IIIb retrograde, the false lumens of the descending and abdominal aortas were more frequently totally thrombosed and the true lumen area of the descending aorta was significantly larger in the high-score group, although no significant difference in the surgical procedure was noted. In the current study, the aforementioned points 1) and 2) could be the most important findings supporting our hypothesis.

Blumenthal et al. studied the relationship between age and the amounts of calcium in the intima and media of 540 human aortic specimens and found that calcifications were more common in the tunica media than in the intima at all ages [29]. Moreover, pathological examinations have indicated that aortic dissection initially develops in the tunica media [9]. Therefore, aortic calcifications present between the intima and the tunica media may prevent separation of these aortic wall layers and consequently reduce progression of the aortic dissection. Based on an in vitro study, vascular smooth muscle cell in the tunica media regulates vascular microcalcification via several miRNAs and modulates vascular remodeling [30, 31]. Vascular calcification is strongly associated with elastin degradation and smooth muscle cell phenotypic change [32, 33]. This indicates that aortic calcifications change the biomechanical properties of the tunica media in the aorta and influence the true and false lumen dimension and the distal extent of ATAAD in the high-score group.

In preoperative CTA, aortic dissection was significantly less distally extended in the high-score group than in the low-score group. Accordingly, DeBakey type II and false lumen thrombosis were more frequently observed in the high-score group than in the low-score group. DeBakey type II dissections were correlated with atherosclerotic disease, and the prevalence of distally extended aortic dissection was lower in patients with non-communicating false lumens than in those with patent false lumens, which are consistent with the results of previous studies [34, 35]. Furthermore, there is a greater decrease in false lumen pressure in aortic dissection with a thrombosed false lumen compared with a patent false lumen [36]. A false lumen thrombosis in the high-score group might cause a decrease in the false lumen pressure and slow the disease progression. In fact, we found that the true lumen/total lumen area ratios of the ascending and descending aorta were likely to be larger in the high-score group than those in the low-score group on preoperative CTA. These findings suggest that the ratio of the true lumen pressure to the false lumen pressure in the high-score group could be higher than that in the low-score group.

Intriguingly, in the early postoperative CTA, the true lumen area of the descending aorta was significantly larger and false lumen thrombosis was significantly more frequently observed in the high-score group. Moreover, changes in the distal extent score and true and false lumen area of the descending aorta from pre- to post-surgery did not significantly differ between the groups. Hence, a lower disease progression in the high-score group before surgery could be consistent with that after surgery. Interestingly, dilatation of the residual dissected descending aorta was smaller in the high-score group than in the low-score group 6 months after surgery. Several imaging findings can help predict the course of residual dissected aorta remodeling after surgery for ATAAD. Progressive dilatation of the descending aorta, persistent intimal tear at the residual aorta, and refilling from the false lumen of a dissected aortic arch were considered the predictors of failing residual aortic remodeling after surgical repair for ATAAD [37, 38]. The current study showed that the aortic Agatston score could be correlated with ATAAD progression and could help predict postoperative residual dissected descending aorta remodeling. Although further investigation is needed to elucidate the influence of the aortic Agatston score on clinical outcomes in patients who undergo surgical repair for ATAAD, our primary aim was to reveal the association between the aortic Agatston score and disease progression of ATAAD in terms of CTA variables.

The most obvious limitations of this study were its retrospective nature, small number of participants, and single-center design, which are potential sources of bias. This study was further limited by the lack of pathological proof and that we did not specifically determine whether calcifications were located in the intima or in the media of the aortic wall. However, it should be noted that we did not aim to unravel the pathology of these calcifications. Further studies with a larger number of patients and greater emphasis on hemodynamic and biomechanical parameters should be performed to evaluate these associations. These data provide important new insights into the association between aortic calcification and the progression of aortic dissection and potentially contribute to the development of a risk assessment system in the patients with high Agatston scores.

## Conclusions

In conclusion, we found that high aortic Agatston scores were significantly correlated with larger true lumen areas of the ascending and descending aorta and with smaller false lumen areas of the descending aorta in patients with ATAAD. Furthermore, compared with the patients with the low Agatston scores, the patients with the high Agatston scores were more likely to have DeBakey type II dissection and false lumen thrombosis, and their aortic dissection was less distally extended. In the early postoperative CTA, the true lumen area of the descending aorta was significantly larger, and false lumen thrombosis was more frequently observed in patients with high Agatston scores, which were consistent with preoperative findings. The aortic Agatston scores could be correlated with ATAAD progression and could help predict postoperative residual dissected descending aorta remodeling.

## Supporting information

**S1 Fig. Regression tree for aortic Agatston score thresholds predictive of distal extent score in CART analysis.**
(TIFF)

**S1 Table. CTA variables of descending aorta before and after surgery in the patients with DeBakey I or IIIb retrograde.**
(DOCX)

**S2 Table. Early Postoperative changes of CTA variables in the patients with DeBakey I or IIIb retrograde (before—after surgery).**
(DOCX)

**S3 Table. Mid-term Postoperative changes of plain CTA variables in the patients with DeBakey I or IIIb retrograde in 6 months after surgery.**
(DOCX)

## Acknowledgments

The authors would like to thank Enago (www.enago.jp) for the English language review.

## Author Contributions

**Conceptualization:** Yasushi Tashima, Shinichi Iwakoshi, Takeshi Inoue, Taichi Sano, Naoyuki Kimura, Takashi Inoue, Koichi Adachi, Atsushi Yamaguchi.

**Data curation:** Yasushi Tashima, Shinichi Iwakoshi, Takeshi Inoue, Noriyuki Nakamura, Taichi Sano, Takashi Inoue, Atsushi Yamaguchi.

**Formal analysis:** Yasushi Tashima, Shinichi Iwakoshi, Takeshi Inoue, Noriyuki Nakamura, Taichi Sano, Naoyuki Kimura, Takashi Inoue, Koichi Adachi.

**Investigation:** Yasushi Tashima, Shinichi Iwakoshi, Takeshi Inoue, Noriyuki Nakamura, Taichi Sano, Naoyuki Kimura, Takashi Inoue, Koichi Adachi, Atsushi Yamaguchi.

**Methodology:** Yasushi Tashima, Shinichi Iwakoshi, Takeshi Inoue, Noriyuki Nakamura, Taichi Sano, Naoyuki Kimura, Takashi Inoue, Koichi Adachi, Atsushi Yamaguchi.

**Project administration:** Yasushi Tashima, Shinichi Iwakoshi, Takeshi Inoue, Naoyuki Kimura.

**Resources:** Yasushi Tashima, Shinichi Iwakoshi, Takeshi Inoue, Noriyuki Nakamura, Taichi Sano, Naoyuki Kimura, Takashi Inoue, Koichi Adachi, Atsushi Yamaguchi.

**Software:** Yasushi Tashima, Shinichi Iwakoshi, Takeshi Inoue, Noriyuki Nakamura, Taichi Sano, Naoyuki Kimura, Takashi Inoue, Koichi Adachi, Atsushi Yamaguchi.

**Supervision:** Yasushi Tashima, Shinichi Iwakoshi, Takeshi Inoue, Noriyuki Nakamura, Taichi Sano, Naoyuki Kimura, Takashi Inoue, Koichi Adachi, Atsushi Yamaguchi.

**Validation:** Yasushi Tashima, Shinichi Iwakoshi, Takeshi Inoue, Noriyuki Nakamura, Taichi Sano, Naoyuki Kimura, Takashi Inoue, Koichi Adachi, Atsushi Yamaguchi.

**Visualization:** Yasushi Tashima, Shinichi Iwakoshi, Takeshi Inoue, Noriyuki Nakamura, Taichi Sano, Naoyuki Kimura, Takashi Inoue, Koichi Adachi, Atsushi Yamaguchi.

**Writing – original draft:** Yasushi Tashima, Shinichi Iwakoshi, Takeshi Inoue.

**Writing – review & editing:** Yasushi Tashima, Shinichi Iwakoshi, Noriyuki Nakamura, Taichi Sano, Naoyuki Kimura.

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
