## [Decision Letter · Decision Letter 0]

15 Oct 2021

PONE-D-21-14950Aortic Agatston score correlates with the progression of acute type A aortic dissectionPLOS ONE

Dear Dr. Tashima

Thank you for submitting your manuscript to PLOS ONE. After careful consideration, we feel that it has merit but does not fully meet PLOS ONE’s publication criteria as it currently stands. Therefore, we invite you to submit a revised version of the manuscript that addresses the points raised during the review process.

We look forward to receiving your revised manuscript.

Kind regards,

Xianwu Cheng, M.D., Ph.D., FAHA

Academic Editor

PLOS ONE

Journal Requirements:

2. Thank you for providing the name of your Institutional Review Board and approval number in your Ethics Statement. We ask that you also provide this information in your Methods section.

Additional Editor Comments (if provided):

Although the topic is interesting, two reviewers have concerned some size as preliminary study or/and discussion and conclusions (over-talk and overreached statement). In addition, the authors should provide sufficient detail for nay other researcher to reproduces the study. Therefore, in the methods sections, the authors describe in detail including the criteria for different patient conditions (hypertension, diabetes, dyslipidemia etc.) and excluding criteria.

Reviewers' comments:

Reviewer's Responses to Questions

**Comments to the Author**

1. Is the manuscript technically sound, and do the data support the conclusions?

Reviewer #1: Partly

Reviewer #2: Yes

2. Has the statistical analysis been performed appropriately and rigorously? 

Reviewer #1: No

Reviewer #2: Yes

3. Have the authors made all data underlying the findings in their manuscript fully available?

Reviewer #1: No

Reviewer #2: Yes

4. Is the manuscript presented in an intelligible fashion and written in standard English?

Reviewer #1: Yes

Reviewer #2: Yes

5. Review Comments to the Author

Reviewer #1: In this study, the authors aimed to explore the association of aortic Agatston scores with disease progression in patients with ATAAD. This was a retrospective analysis of 85 patients who had preoperatively undergone CTA for evaluation of aortic calcification. The patients were classified into high score (Agatston score ≥ 3344; n = 49) and low-score (<3344; n = 36) groups. Correlations between aortic Agatston scores and CTA variables were assessed. Higher aortic Agatston scores were significantly correlated with the smaller distal extent of aortic dissection (p <0.001), larger true lumen areas of the ascending (p = 0.009) and descending aorta (p = 0.002), and smaller false lumen areas of the descending aorta (p = 0.028). Patients in the high-score group were more likely to have DeBakey type II dissection (p = 0.001) and false lumen thrombosis (p = 0.027) than those in the low-score group, thereby confirming the correlations. Aortic dissection in the high score group was significantly less distally extended (p < 0.001). The authors concluded that higher aortic Agatston score correlates with the larger true lumen area of the ascending and descending aorta and the less distal progression of aortic dissection in patients with ATAAD.

This manuscript is engaging and thought-provoking. I present the following suggestions / comments in the hopes of improving the manuscript. At this current stage, I recommend major revision for publication in PlosOne.

Major Comments

1. Introduction: Line 69 represents a transition point, and therefore, I recommend a new paragraph that guides the reader more concisely to purpose of this investigation. In line 67, instead of “we hypothesize”, I recommend “It is plausible aortic calcification could alter the biomechanical…”

2. The follow up study time frame of 7 days is too abbreviated to evaluate early aortic progression especially since it is not clear if the extent of dissection changed from pre- and post-op CTA (Norton et al., 2020, J Thorac Cardiovasc Surg, PMID: 32517536). This lack of clarification is a major limitation of the current manuscript version. It seems more appropriate for this study to evaluate the association of aortic calcification and extent of aortic dissection and other properties of the aorta.

3. Methods: Line 88-89 should be revised to examine the association of aortic calcification with the extent of aortic dissection.

4. Results: The extent of dissection from pre- to post-op needs to be evaluated; otherwise, the reader does not know if the outcome changed.

5. Discussion: This manuscript has a small sample size with the possibility of having a straightforward objective and conclusion. The sample size is a limitation, but this is overcome if the authors remain focused on the central findings, which from my view, is the potential clinical utility of Agatston score to evaluate extent of dissection and other aortic properties.

I recommend rewriting the discussion, starting with deleting the introductory paragraph of Discussion as it repeats to the Introduction and starting with “This study demonstrates…points 1-3” The subsequent paragraph should follow points 1-3 with supportive evidence. Of points 1-3, which is the most clinically important finding? Start the discussion addressing this point and then build in the other discussion points. The conclusion paragraph reiterates the results, but there is conclusion statement that tells the reader the translational importance of this work. Reworking the discussion will strengthen this manuscript.

6. Abstract: Please rewrite after revising the analysis.

Reviewer #2: The authors provide an interesting and potential important manuscript describing "Aortic Agatston score correlates with the progression of acute type A aortic dissection", The main issues concerning this paper are those concerning the potential associations between Aortic Agatston score and type A aortic dissection.

There are some weak points that need to be addressed by the authors

Major

1. Why Agatston Score 3344 is selected as the thresholds between the high and low scores of ATAAD.

2. Some recent literature on arterial dissection and calcification needs to be cited

6. PLOS authors have the option to publish the peer review history of their article (what does this mean?). If published, this will include your full peer review and any attached files.

Reviewer #1: No

Reviewer #2: No

---

## [Author Response · Author response to Decision Letter 0]

19 Nov 2021

November 27, 2021 

Dr. Emily Chenette

Editor-in-Chief

PLOS ONE

Manuscript ID: PONE-D-21-14950

Aortic Agatston score correlates with the progression of acute type A aortic dissection 

Dear Editor: 

Please find enclosed the responses to the reviewers’ comments. Thank you for providing us the opportunity to revise the manuscript. The revisions, both marked and unmarked, are included. Our team will be honored if the manuscript is published in PLOS ONE.

Reviewer #1: In this study, the authors aimed to explore the association of aortic Agatston scores with disease progression in patients with ATAAD. This was a retrospective analysis of 85 patients who had preoperatively undergone CTA for evaluation of aortic calcification. The patients were classified into high score (Agatston score ≥ 3344; n = 49) and low-score (<3344; n = 36) groups. Correlations between aortic Agatston scores and CTA variables were assessed. Higher aortic Agatston scores were significantly correlated with the smaller distal extent of aortic dissection (p <0.001), larger true lumen areas of the ascending (p = 0.009) and descending aorta (p = 0.002), and smaller false lumen areas of the descending aorta (p = 0.028). Patients in the high-score group were more likely to have DeBakey type II dissection (p = 0.001) and false lumen thrombosis (p = 0.027) than those in the low-score group, thereby confirming the correlations. Aortic dissection in the high score group was significantly less distally extended (p < 0.001). The authors concluded that higher aortic Agatston score correlates with the larger true lumen area of the ascending and descending aorta and the less distal progression of aortic dissection in patients with ATAAD.

This manuscript is engaging and thought-provoking. I present the following suggestions / comments in the hopes of improving the manuscript. At this current stage, I recommend major revision for publication in PlosOne.

Major Comments

1. Introduction: Line 69 represents a transition point, and therefore, I recommend a new paragraph that guides the reader more concisely to purpose of this investigation. In line 67, instead of “we hypothesize”, I recommend “It is plausible aortic calcification could alter the biomechanical…”

→Thank you very much for the suggestion. We have revised the statement to “It is plausible that aortic calcifications could alter the biomechanical properties of the aorta in patients with acute type A aortic dissection (ATAAD)” (page 3, lines 69-71).

2. The follow up study time frame of 7 days is too abbreviated to evaluate early aortic progression especially since it is not clear if the extent of dissection changed from pre- and post-op CTA (Norton et al., 2020, J Thorac Cardiovasc Surg, PMID: 32517536). This lack of clarification is a major limitation of the current manuscript version. It seems more appropriate for this study to evaluate the association of aortic calcification and extent of aortic dissection and other properties of the aorta.

→Thank you for the insightful comment.

In CTA within 7 days after surgery, the true lumen area of the descending aorta was significantly larger in the high-score group than in the low-score group. Further, the false lumen area of the descending aorta was significantly smaller in the high-score group than in the low-score group. 

However, as you suggested, we must validate the change from pre-surgery to early post-surgery. We assessed changes in CTA variables from pre- to post-surgery in each group (S1 Table A) and differences in changes between groups (S1 Table B). Compared with preoperative CTA variables, the postoperative true lumen dimension of the descending aorta was small in the high- and low-score groups (p = 0.038, 0.058, respectively). However, the distal extent score after surgery did not significantly change (S1 Table A). Moreover, there was no remarkable difference in the changes in the distal extent score and the true and false lumen area of the descending aorta from pre-surgery to early post-surgery between the two groups (S1 Table B). Therefore, the pre- and postoperative results were consistent.

With a smaller distal extent score, the prevalence of false lumen thrombosis was higher, and larger true lumen area in the descending aorta in the high-score group could be correlated with midterm residual dissected descending aorta remodeling after surgery1, 2. Moreover, we evaluated the midterm changes in descending aortic diameter and area after surgery between the groups. We collected data about descending aortic diameter and area via plain CT scan 6 months after surgery and assessed changes 6 months after surgery between groups. 

Interestingly, dilatation of the descending aortic diameter and area 6 months after surgery was smaller in the high-score group than in the low-score group (Des diameter change in fold change/CT variables 6 months after surgery – high-score group: 0.98 [0.87–1.04] vs. low-score group: 1.01 [0.94–1.1], p=0.058; Des area change 7 days after surgery – 1.03 [0.88–1.13] vs. 1.11 [0.97–1.29], p=0.031) (S2 Table). Although other clinical factors should be adjusted to compare midterm descending aorta remodeling, these findings suggested that the aortic Agatston score could be an important factor for predicting descending aortic dilatation after surgery.

The following sentences were added in the Methods and Results sections.

“Postoperative CTA was performed within 7 days after surgery to assess early aortic dissection progression after the initial surgery and changes in CTA variables from pre- to post-surgery. Furthermore, postoperative plain computed tomography (CT) scan was conducted 6 months after surgery to assess midterm changes in descending aortic dimension and diameter after surgery between the two groups.” (page 5. lines 104–108).

“Compared with preoperative CTA variables, the postoperative true lumen dimension of the descending aorta was small in the high- and low-score groups (p = 0.038, 0.058, respectively). However, the distal extent score after surgery did not significantly change (S1 Table A). Moreover, there was no remarkable difference in the changes in the distal extent score and true and false lumen area of the descending aorta from pre- to post-surgery between the two groups (S1 Table B). Hence, high aortic Agatston scores could be correlated with a slower progression of residual dissected descending aorta before and after surgery.

Moreover, we assessed and compared the midterm changes in descending aortic diameter and area via plain CT scan 6 months after surgery and CT variables within 7 days after surgery. Interestingly, dilatation of the descending aortic diameter and area 6 months after surgery were smaller in the high-score group than in the low-score group (Des diameter change: 0.98 [0.87–1.04] vs. 1.01 [0.94–1.1], p = 0.058; Des area change: 1.03 [0.88–1.13] vs. 1.11 [0.97–1.29], p = 0.031, in fold change/CT variables within 7days after surgery) (S2 Table).” (page 14, lines 243–255).

3. Methods: Line 88-89 should be revised to examine the association of aortic calcification with the extent of aortic dissection.

→Thank you for the suggestion. 

This statement was revised to “Then, we examined the association between aortic calcifications and the extent of aortic dissection in these patient groups.” (page 4, lines 92–93).

4. Results: The extent of dissection from pre- to post-op needs to be evaluated; otherwise, the reader does not know if the outcome changed.

→Thank you very much. 

The distal extent score before and after surgery did not significantly differ between the two groups, as shown in S1 Table A. The following sentences were added in the Results section:

“Compared with preoperative CTA variables, the postoperative true lumen dimension of the descending aorta was small in the high- and low-score groups (p = 0.038, 0.058, respectively). However, the distal extent score after surgery did not significantly change (S1 Table A)” (page 14, lines 243–246).

5. Discussion: This manuscript has a small sample size with the possibility of having a straightforward objective and conclusion. The sample size is a limitation, but this is overcome if the authors remain focused on the central findings, which from my view, is the potential clinical utility of Agatston score to evaluate extent of dissection and other aortic properties.

I recommend rewriting the discussion, starting with deleting the introductory paragraph of Discussion as it repeats to the Introduction and starting with “This study demonstrates…points 1-3” The subsequent paragraph should follow points 1-3 with supportive evidence. Of points 1-3, which is the most clinically important finding? Start the discussion addressing this point and then build in the other discussion points. The conclusion paragraph reiterates the results, but there is conclusion statement that tells the reader the translational importance of this work. Reworking the discussion will strengthen this manuscript.

Thank you for the advice.

We deleted the introductory paragraph and started the Discussion with the findings. To validate which is the most important finding in the study, the following sentences were added in the Discussion section:

“In the current study, the aforementioned points 1) and 2) could be the most important findings supporting our hypothesis.” (page 16, lines 270–271).

We added the following data from new references to explain that aortic calcification could be an important factor for changes in the biomechanical property of the tunica media in vitro and could influence ATAAD progression.

“Based on an in vitro study, vascular smooth muscle cell in the tunica media regulates vascular microcalcification via several miRNAs and modulates vascular remodeling [30, 31]. Vascular calcification is strongly associated with elastin degradation and smooth muscle cell phenotypic change [32, 33]. This indicates that aortic calcifications change the biomechanical properties of the tunica media in the aorta and influence the true and false lumen dimension and the distal extent of ATAAD in the high-score group.” (page 16, lines 277–283).

30. Badi I, Mancinelli L, Polizzotto A, Ferri D, Zeni F, Burba I, Milano G, Brambilla F, Saccu C, Bianchi ME, Pompilio G, Capogrossi MC and Raucci A. miR-34a Promotes Vascular Smooth Muscle Cell Calcification by Downregulating SIRT1 (Sirtuin 1) and Axl (AXL Receptor Tyrosine Kinase). Arterioscler Thromb Vasc Biol. 2018;38:2079-2090.

31. Shi N, Mei X and Chen SY. Smooth Muscle Cells in Vascular Remodeling. Arterioscler Thromb Vasc Biol. 2019;39:e247-e252.

32. Pai A, Leaf EM, El-Abbadi M and Giachelli CM. Elastin degradation and vascular smooth muscle cell phenotype change precede cell loss and arterial medial calcification in a uremic mouse model of chronic kidney disease. Am J Pathol. 2011;178:764-73.

33. Bhat OM, Yuan X, Cain C, Salloum FN and Li PL. Medial calcification in the arterial wall of smooth muscle cell-specific Smpd1 transgenic mice: A ceramide-mediated vasculopathy. J Cell Mol Med. 2020;24:539-553.

The following data based on new references were added to provide evidence that DeBakey type II dissections was correlated with atherosclerotic disease, which is consistent with our findings.

“DeBakey type II dissections were correlated with atherosclerotic disease, and the prevalence of distally extended aortic dissection was lower in patients with non-communicating false lumens than in those with patent false lumens, which are consistent with the results of previous studies [34, 35].” (page 16, lines 285–288).

34. Akutsu K, Yoshino H, Tobaru T, Hagiya K, Watanabe Y, Tanaka K, Koyama N, Yamamoto T, Nagao K and Takayama M. Acute type B aortic dissection with communicating vs. non-communicating false lumen. Circ J. 2015;79:567-73.

35. Philip JL, De Oliveira NC, Akhter SA, Rademacher BL, Goodavish CB, DiMusto PD and Tang PC. Cluster analysis of acute ascending aortic dissection provides novel insight into mechanisms of distal progression. J Thorac Dis. 2017;9:2966-2973.

The following data from new references were added to explain postoperative findings and confirm whether the aortic Agatston score can predict postoperative residual dissected descending aorta remodeling.

“Moreover, changes in the distal extent score and true and false lumen area of the descending aorta from pre- to post-surgery did not significantly differ between the groups. Hence, a lower disease progression in the high-score group before surgery could be consistent with that after surgery. Interestingly, dilatation of the residual dissected descending aorta was smaller in the high-score group than in the low-score group 6 months after surgery. Several imaging findings can help predict the course of residual dissected aorta remodeling after surgery for ATAAD. Progressive dilatation of the descending aorta, persistent intimal tear at the residual aorta, and refilling from the false lumen of a dissected aortic arch were considered the predictors of failing residual aortic remodeling after surgical repair for ATAAD [37, 38]. The current study showed that the aortic Agatston score could be correlated with ATAAD progression and could help predict postoperative residual dissected descending aorta remodeling.” (page 17, lines 298–308).

37. Leontyev S, Haag F, Davierwala PM, Lehmkuhl L, Borger MA, Etz CD, Misfeld M, Gutberlet M and Mohr FW. Postoperative Changes in the Distal Residual Aorta after Surgery for Acute Type A Aortic Dissection: Impact of False Lumen Patency and Size of Descending Aorta. Thorac Cardiovasc Surg. 2017;65:90-98.

38. Saremi F, Hassani C, Lin LM, Lee C, Wilcox AG, Fleischman F and Cunningham MJ. Image Predictors of Treatment Outcome after Thoracic Aortic Dissection Repair. Radiographics. 2018;38:1949-1972.

The following sentences were added in the Conclusion section.

“In the early postoperative CTA, the true lumen area of the descending aorta was significantly larger, and false lumen thrombosis was more frequently observed in patients with high Agatston scores, which were consistent with preoperative findings. The aortic Agatston scores could be correlated with ATAAD progression and could help predict postoperative residual dissected descending aorta remodeling.” (page 18, lines 327–332).

6. Abstract: Please rewrite after revising the analysis.

→Thank you very much. The abstract was modified by adding new analysis results.

Reviewer #2: The authors provide an interesting and potential important manuscript describing "Aortic Agatston score correlates with the progression of acute type A aortic dissection", The main issues concerning this paper are those concerning the potential associations between Aortic Agatston score and type A aortic dissection.

There are some weak points that need to be addressed by the authors

Major

1. Why Agatston Score 3344 is selected as the thresholds between the high and low scores of ATAAD.

→Thank you very much for this question.

We used the classification and regression tree (CART), a machine-learning algorithm for clinical decision-making that can be used to determine the breakpoint and identify aortic Agatston score thresholds predictive of disease progression (distal extent of aortic dissection) in patients with ATAAD. CART is a nonparametric decision tree learning technique that produces either classification or regression trees based on whether the dependent variable is categorical or numeric, respectively [3].

The current study found that the distal extent score of ATAAD was strongly correlated with aortic Agatston score (rho = −0.446, 95% CI: −0.602 to −0.257, p value < 0.001), as shown in Table 2. According to Reference [4], the statistician established the regression tree to identify aortic Agatston score thresholds predictive of distal extent score with R software (Saitama Medical Center, Jichi Medical University, Saitama, Japan) (S1 Fig). The threshold of aortic Agatston score was 3344. In fact, we confirmed that the distal extent score of patients with an aortic Agatston score of ≥ 3344 was significantly lower than that of patients with an aortic Agatston score of < 3344 (5 [0–8] vs. 10 [7–11], p < 0.001), as shown in Table 3.

We added sentences with references and a supplemental figure in the Methods section.

“CART is a nonparametric decision tree learning technique that produces either classification or regression trees based on whether the dependent variable is categorical or numeric, respectively [25, 26].” (page 4. lines 87–89).

25. L. Breiman, J.H. Friedman, R.A. Olshen, and C.J Stone, "Classification and Regression Trees," Wadsworth, Belmont, Ca, 1983.

26. Terry M. Therneau, Elizabeth J. Atkinson, Mayo Foundation, "An Introduction to Recursive Partitioning Using the RPART Routines," https://rdrr.io/cran/rpart/f/ inst/doc/longintro.pdf, April 11, 2019.

S1 Fig. Regression tree for aortic Agatston score thresholds predictive of distal extent score in CART analysis

2. Some recent literature on arterial dissection and calcification needs to be cited

→Thank you very much. 

Only few reports have investigated the association between aortic calcifications and aortic dissection [12-14].

12. Akiyoshi K, Kimura N, Aizawa K, Hori D, Okamura H, Morita H, Adachi K, Yuri K, Kawahito K and Yamaguchi A. Surgical outcomes of acute type A aortic dissection in dialysis patients. Gen Thorac Cardiovasc Surg. 2019;67:501-509.

13. Yang CJ, Tsai SH, Wang JC, Chang WC, Lin CY, Tang ZC and Hsu HH. Association between acute aortic dissection and the distribution of aortic calcification. PLoS One. 2019;14:e0219461.

14. de Jong PA, Hellings WE, Takx RA, Isgum I, van Herwaarden JA and Mali WP. Computed tomography of aortic wall calcifications in aortic dissection patients. PLoS One. 2014;9:e102036.

We added new references explaining that aortic calcification could be an important factor for managing the biomechanical property of tunica media in vitro study and could influence ATAAD progression [30-33].

30. Badi I, Mancinelli L, Polizzotto A, Ferri D, Zeni F, Burba I, Milano G, Brambilla F, Saccu C, Bianchi ME, Pompilio G, Capogrossi MC and Raucci A. miR-34a Promotes Vascular Smooth Muscle Cell Calcification by Downregulating SIRT1 (Sirtuin 1) and Axl (AXL Receptor Tyrosine Kinase). Arterioscler Thromb Vasc Biol. 2018;38:2079-2090.

31. Shi N, Mei X and Chen SY. Smooth Muscle Cells in Vascular Remodeling. Arterioscler Thromb Vasc Biol. 2019;39:e247-e252.

32. Pai A, Leaf EM, El-Abbadi M and Giachelli CM. Elastin degradation and vascular smooth muscle cell phenotype change precede cell loss and arterial medial calcification in a uremic mouse model of chronic kidney disease. Am J Pathol. 2011;178:764-73.

33. Bhat OM, Yuan X, Cain C, Salloum FN and Li PL. Medial calcification in the arterial wall of smooth muscle cell-specific Smpd1 transgenic mice: A ceramide-mediated vasculopathy. J Cell Mol Med. 2020;24:539-553.

The following references were added to provide evidence showing that DeBakey type III dissections were correlated with atherosclerotic disease [35] and to explain the predictor of postoperative residual dissected descending aorta remodeling [37, 38]. 

35. Philip JL, De Oliveira NC, Akhter SA, Rademacher BL, Goodavish CB, DiMusto PD and Tang PC. Cluster analysis of acute ascending aortic dissection provides novel insight into mechanisms of distal progression. J Thorac Dis. 2017;9:2966-2973.

37. Leontyev S, Haag F, Davierwala PM, Lehmkuhl L, Borger MA, Etz CD, Misfeld M, Gutberlet M and Mohr FW. Postoperative Changes in the Distal Residual Aorta after Surgery for Acute Type A Aortic Dissection: Impact of False Lumen Patency and Size of Descending Aorta. Thorac Cardiovasc Surg. 2017;65:90-98.

38. Saremi F, Hassani C, Lin LM, Lee C, Wilcox AG, Fleischman F and Cunningham MJ. Image Predictors of Treatment Outcome after Thoracic Aortic Dissection Repair. Radiographics. 2018;38:1949-1972.

References

1. Leontyev S, Haag F, Davierwala PM, Lehmkuhl L, Borger MA, Etz CD, Misfeld M, Gutberlet M and Mohr FW. Postoperative Changes in the Distal Residual Aorta after Surgery for Acute Type A Aortic Dissection: Impact of False Lumen Patency and Size of Descending Aorta. Thorac Cardiovasc Surg. 2017;65:90-98.

2. Saremi F, Hassani C, Lin LM, Lee C, Wilcox AG, Fleischman F and Cunningham MJ. Image Predictors of Treatment Outcome after Thoracic Aortic Dissection Repair. Radiographics. 2018;38:1949-1972.

3. L. Breiman, J.H. Friedman, R.A. Olshen, and C.J Stone, "Classification and Regression Trees," Wadsworth, Belmont, Ca, 1983.

4. Terry M. Therneau, Elizabeth J. Atkinson, Mayo Foundation, "An Introduction to Recursive Partitioning Using the RPART Routines," https://rdrr.io/cran/rpart/f/ inst/doc/longintro.pdf, April 11, 2019.

---

## [Decision Letter · Decision Letter 1]

31 Jan 2022

Aortic Agatston score correlates with the progression of acute type A aortic dissection

PONE-D-21-14950R1

Dear Dr Dashima

We’re pleased to inform you that your manuscript has been judged scientifically suitable for publication and will be formally accepted for publication once it meets all outstanding technical requirements.

Kind regards,

Xianwu Cheng, M.D., Ph.D., FAHA

Academic Editor

PLOS ONE

Additional Editor Comments (optional):

Although the original reviewer has declined to review the revised manuscript, all of original concerns have been addressed by the authors.

Reviewers' comments:

Reviewer's Responses to Questions

**Comments to the Author**

1. If the authors have adequately addressed your comments raised in a previous round of review and you feel that this manuscript is now acceptable for publication, you may indicate that here to bypass the “Comments to the Author” section, enter your conflict of interest statement in the “Confidential to Editor” section, and submit your "Accept" recommendation.

Reviewer #2: All comments have been addressed

2. Is the manuscript technically sound, and do the data support the conclusions?

Reviewer #2: Yes

3. Has the statistical analysis been performed appropriately and rigorously? 

Reviewer #2: Yes

4. Have the authors made all data underlying the findings in their manuscript fully available?

Reviewer #2: Yes

5. Is the manuscript presented in an intelligible fashion and written in standard English?

Reviewer #2: Yes

6. Review Comments to the Author

Reviewer #2: (No Response)

7. PLOS authors have the option to publish the peer review history of their article (what does this mean?). If published, this will include your full peer review and any attached files.

Reviewer #2: No

---

## [Editor Report · Acceptance letter]

3 Feb 2022

PONE-D-21-14950R1 

Aortic Agatston score correlates with the progression of acute type A aortic dissection 

Dear Dr. Tashima:

I'm pleased to inform you that your manuscript has been deemed suitable for publication in PLOS ONE. Congratulations! Your manuscript is now with our production department. 

Kind regards, 

on behalf of

Associate Prof. Xianwu Cheng 

Academic Editor

PLOS ONE